# Examining the Mental Well-Being of Australian Sport Coaches

**DOI:** 10.3390/ijerph16234601

**Published:** 2019-11-20

**Authors:** Fraser Carson, Mary Malakellis, Julia Walsh, Luana C. Main, Peter Kremer

**Affiliations:** 1Centre for Sport Research, School of Exercise and Nutrition Sciences, Deakin University, Geelong, VIC 3220, Australia; mary.malakellis1@deakin.edu.au (M.M.); jwalsh56@me.com (J.W.); peter.kremer@deakin.edu.au (P.K.); 2Institute for Physical Activity and Nutrition, School of Exercise and Nutrition Sciences, Deakin University, Geelong, VIC 3220, Australia; luana.main@deakin.edu.au

**Keywords:** workload, control, areas of work life, stress, self-regulation

## Abstract

*Background:* Research has highlighted the multitude of factors that are negatively associated with coach mental well-being but has failed to investigate how the determinants of mental well-being can affect the coach both positively and negatively. Accordingly, the aim of this study was to investigate levels of mental well-being among sport coaches and assess whether areas of work life—specifically workload and control—are related to levels of mental well-being. *Method:* An online survey comprising demographic and coaching experience details, the Areas of Work Life Scale (AWS), and the Warwick–Edinburgh Mental Well-Being Scale was completed by 464 Australian coaches involved in a range of sports. Differences in coach mental well-being according to key demographic and coaching-related subgroups were assessed using separate t-test and ANOVA analyses and the magnitude of effects was determined using Cohen’s d and the eta-squared (*ή*^2^) statistics. Multiple linear regression was used to examine relationships between both workload and control and mental well-being after controlling for age, gender, coaching setting and weekly coaching activity. *Results:* The findings indicate poorer mental well-being among both male and younger coaches and indicate that coach mental well-being is related to the ability to self-manage the workload associated with their role as a coach as well as greater autonomy over coaching-related tasks and activities. Specifically, a one-unit increase in AWS workload and AWS control were associated with ~three- and ~four-unit increases in coach mental well-being, respectively. *Conclusion*: Greater provision of resources and education is required to assist coaches to manage their own mental well-being, while being supported by the organisation they coach for. Enabling coaches to balance their coaching requirements and to have control over their environment will improve their ability to constantly coach at a high standard.

## 1. Introduction

Coaches, at all levels of sport, experience many stressors (i.e., athlete performance, expectations, external scrutiny) related to their daily coaching activity [1]. Given the nature of the coaching role and the dynamic nature of the sport context in which they operate, this repeated exposure to stressors has the potential to impact their mental well-being. Mental well-being has been defined as a person’s psychological functioning, satisfaction with life and their ability to create/maintain mutually beneficial relationships [2]. Higher levels of mental well-being (i.e., above 59 on the Warwick–Edinburgh Mental Well-Being Scale (WEMWBS)) are associated with more happiness, subjective vitality and positivity [3], while lower levels (i.e., below 40 on the WEMWBS) are related to emotional and physical exhaustion, low satisfaction, and a lack of caring [4]. Research demonstrating links between workplace stressors and levels of employee mental well-being comes from a range of workplace settings [5]. However, few studies have examined levels of mental well-being, as well as factors associated with mental well-being, among those involved in the sport workforce. This is particularly true for individuals who work or volunteer as coaches.

Across all levels of sport, the coach’s role is to design and deliver a systematic program to facilitate improvement in individual and team performance. Coaches are also required to provide a safe environment and limit potential risks for participants. At the community level (i.e., local/community sporting competitions), coaches often have a variety of roles and responsibilities including youth engagement, dealing with parents, organisational fundraising, and multiple administrative duties [6]. In high-performance sport (i.e., national/international level competition), they are influential in the development of the organisational culture, building brand awareness, and are an integral face of the organisation [7]. High-performance coaches also maintain long work hours, have short-term contracts and low job security [8], while under constant scrutiny from media and public. To effectively manage these interactions, the coach must be able to maintain high levels of mental well-being and functioning [9].

Normative levels of mental well-being have been reported in Australia’s general population (M = 48.1) [10] and within specific populations (i.e., teachers, M = 47.2 [11]; general practitioners, M = 50.2, [12]) using the WEMWBS. These studies have also identified a number of important correlates of mental well-being. In brief, these include personality, financial stability, employment type [13], cognitive and emotional demands [14], opportunity to develop, and positive social relationships [15]. An inability of organisations to provide flexible working arrangements also impacts employee mental well-being negatively [5]. Having meaningful employment increases mental well-being [16], with job satisfaction associated with reduced potential for burnout [17]. Other studies have reported higher levels of mental well-being associated with improved psychological functioning (i.e., an ability to achieve personal goals) as individuals age, particularly the use of more adaptive coping strategies [18]. Levels of mental well-being change over time, with autonomy and competence increasing with age [19]. Examining job-related stressors, Dewe, O’Driscoll and Cooper [5] noted workload, poor communication, and job security were among the key concerns of employees. Arnold and Fletcher’s [20] research synthesis found 640 distinct stressors in the sport workplace, with organisational leadership and culture issues of prime importance. They identified that many employees in sports organisations often have unclear or unrealistic goals, with no control over these.

According to the social-ecological model [21] a number of personal (i.e., age; gender) and coaching (i.e., competition setting) variables are associated with coach mental well-being. Older coaches are better able to self-regulate against stressors than younger coaches [22]. Coaches with less ability to manage workload are more susceptible to poorer mental well-being [23], while coaching in an autonomy-supportive environment can improve mental well-being [24]. McNeill, Durand-Bush and Lemyre [25] found coaches who were at risk of burnout when working on average nine hours longer per week than coaches who were thriving. Kilo and Hassmén’s [26] study of Australian sport coaches recognised negative effects on mental well-being as a result of spending more hours per week coaching and having a higher perceived workload. Job role and control of support staff [27] and conflict with employers [28] are significant stressors for coaches. Norman and Rankin-Wright [29] identified lower levels of mental well-being for female coaches, as they experience poorer work–life balance, fewer social networks in the industry, and lacked trust from their organisation. Older women in coaching also tend to feel undervalued and less supported by their organisation [29].

A general belief is that coaches in higher levels of sports performance experience greater stressors and therefore are more susceptible to lower levels of mental well-being than individuals coaching at the community level. Recent research challenges this notion, with no differences in mental well-being reported by coaches from different performance levels [25] indicating that community-level coaches just experience different stressors (i.e., ensuring athlete continuation in sport) to those coaching in high-performance settings [30]. An inability for coaches to manage workloads leads to great difficulty in managing mental well-being [8]. Higher workloads also predict higher levels of emotional exhaustion [26], which is likely to decrease mental well-being. The level of control a coach has over her/his workplace is an important correlate of mental well-being [31]. Lower levels of control will decrease mental well-being [32].

The aims of this study were to investigate levels of mental well-being among Australian sport coaches and assess whether areas of work life—specifically workload and control—are related to levels of mental well-being. Based on previous literature, it was hypothesised that: (1) sport coaches will report moderate levels of mental well-being and that levels will differ according to individual and coaching factors—specifically coaches who are male, older, and have fewer weekly coaching hours have higher levels of mental well-being; and (2) workload manageability and control will be positively associated with higher levels of mental well-being after controlling for personal (gender, age) and coaching (setting, weekly hours coaching) variables.

## 2. Materials and Methods

### 2.1. Recruitment and Sample

Participants were recruited from sport coaching networks across Australia. A list of 33 national and state sport/coaching organisations were identified and contacted about the study. Representatives from each of the organisations emailed information about the study and associated materials to registered coaches associated with their organisation. The email included a flyer with information about the study, a plain language statement and consent form as well as link to the survey. Collectively, the number of coaches associated with the 33 organisations were estimated at ~4000—from which, 464 completed an online survey. The sample comprised Australian-based coaches involved in coaching (one or more) sports from ‘recreational’ (i.e., community) through to ‘high-performance’ (i.e., elite) levels.

### 2.2. Instrument and Measures

An online survey instrument was used to capture demographic coaching and workplace information as well mental well-being data. Demographic questions included gender and age. Coaching questions included ‘what is your current coach setting?’; ‘what is the highest coach accreditation (e.g., level 1–4) you hold in this sport(s)?’, ‘how many years have you been a coach in this sport (at all levels)?’; ‘do you currently coach a team or individual athletes?’; and ‘total hours of coaching activity during the week (including planning administration, etc.)?’. Work-life experiences was captured using the Areas of Work-life Scale (AWS) [33]. Broadly, the AWS is used to assess work areas that positively or negatively contribute to work engagement and burnout. The AWS consists of 29 items that reflect six dimensions: (1) workload—the ability to manage job demands (5 items; e.g., ‘I have enough time to do what’s important in my job’); (2) control—the perceived capacity of individuals to impact choices identifying their work, to exercise personal autonomy, and to access assets keeping in mind the end goal to finish their work (4 items; e.g., ‘I have control over how I do my work’); (3) reward—the power of reinforcement to shape performance and indicates the degree to which rewards are consistent with the individual’s desires (4 items; e.g., ‘I receive recognition from others for my work’); (4) community—the nature of social interaction that occurs in the workplace (5 items; e.g., ‘People trust one another to fulfill their roles’); (5) fairness—the degree to which choices and asset designation at work are seen as reasonable and respected by others (6 items; e.g., ‘Resources are allocated fairly here’) and (6) values—the standards and inspirations that initially draw individuals to a particular job (4 items; e.g., ‘My values and organisation’s values are alike’). Two of the AWS’ six dimensions—workload and control—were used for the present study. Responses are indicated using a 5-point Likert scale (1 = ‘strongly disagree’, 5 = ‘strongly agree’). Studies have indicated that the AWS is valid and has acceptable reliability (α = 0.78–0.90) [34,35]. Mental well-being was assessed using The Warwick–Edinburgh Mental Wellbeing Scale (WEMWBS) [36]. The WEMWBS comprises 14 statements covering both feelings and functioning aspects of mental well-being based over the previous two weeks [37]; for example, ‘I’ve been feeling close to other people’, ‘I’ve been feeling confident’, and ‘I’ve been feeling relaxed’. Responses are indicated using a 5-point Likert scale (1 = ‘none of the time’, 5 = ‘all of the time’) with higher scores indicating better individual mental well-being. The WEMWBS has good content validity and acceptable internal consistency (α = 0.89) [36] and is an appropriate tool to measure mental health and wellbeing across a range of populations [38].

Coach mental well-being indicated by total WEMWBS score was the primary outcome measure. Key predictors included coach age and gender, coaching setting and weekly coaching activity as well as manageability of workload and work-life control as indicated by the relevant AWS subscales.

### 2.3. Procedure

Participants completed the survey questionnaire anonymously using Qualtrics software (Qualtrics, Provo, UT, USA) during August—November 2017 and typically took approximately 20 minutes. Ethical approval (HEAG-H 97_2017) for the study was provided by the authors’ institutional ethics committee and informed consent obtained from participants.

### 2.4. Analysis

Categorical response and numeric-based free text responses were recoded into ordered categories as follows. Age was recoded into three categories (years; ≤30, 31–50, >50); current coach setting into two categories (high performance and representative = ‘performance’, club and recreational = ‘community’); highest level of coach accreditation for the main sport was coded into four categories (none, level 1 = introductory, level 2 = intermediate, level 3 = advanced—derived by two members of the research team (FC, MM) after reviewing and agreeing on competencies for the different levels across sports indicated by coaches); period coaching their main sport into four categories (years; ≤2, >2–5, >5–10, >10) and weekly coaching activity into three categories (hours; ≤10, >10–20, >20). Descriptive statistics (means, proportions) were used to summarise demographic and coaching variables. Scale and subscale scores were computed for the AWS and WEMWBS according to published protocols and internal consistency assessed using Cronbach’s alpha. Relationships between coach/work-related experiences (AWS workload and control subscales) and coach well-being (WEMWBS scores) were assessed using Pearson’s correlation statistic. Differences in coach well-being according to key demographic and coaching subgroups were assessed using separate t-test and ANOVA analyses and magnitude of effects determined using Cohen’s *d* and the eta-squared (*η*^2^) statistics. Multiple linear regression was used to examine relationships between both workload and control and coach well-being after controlling for age, gender, coaching setting and weekly coaching activity. Analyses were performed using Stata Statistical Software: Release 15 [39] and significance of effects determined as *p* < 0.05.

## 3. Results

Most coaches were male, aged between 31 and 50 years, coached a team sport(s), for less than 20 h/week at the community level. Most coaches held a coach accreditation (level 1 or 2 equivalent) and almost one-third reported having coached in their main sport for more than 10 years (see Table 1).

The internal consistency values for the AWS workload, AWS control and WEMWBS subscales/scales demonstrated were all acceptable; AWS workload α = 0.70, AWS control α = 0.83, and WEMWBS α = 0.92. The AWS workload and control subscales were both positively correlated with the WEMWBS scale (workload: r = 0.35, *p* < 0.001; control: r = 0.42, *p* < 0.001).

The mean well-being score for all coaches was 50.9 (95% confidence interval: 50.2–51.7) indicating that coaches in this sample generally reported high levels of well-being. Mean well-being scores for the demographic and coaching subgroups are presented in Table 2. Higher levels of well-being were reported by older coaches and coaches who reported higher levels of workload manageability as well as higher levels of control over their work life. No differences in coach well-being were observed for gender, coach setting and level of (weekly) coaching activity (see Table 2).

The relationship between personal and coaching workload manageability and control over their work life and mental well-being was examined using multivariate regression. Results of this adjusted analysis are presented in Table 3 and show lower mental well-being scores for male coaches (~2 mental well-being units), and younger- and middle-aged coaches (each ~3 mental well-being units lower) than their female and older-aged counterparts. Coach setting and level of weekly coaching activity were not related to coach mental well-being. Both AWS workload and AWS control variables were significantly related to mental well-being, whereby a one-unit increase in AWS workload and AWS control were associated with ~3 and ~4 unit increases in coach mental well-being (see Table 3).

## 4. Discussion

The primary aim of this study was to investigate mental well-being among sports coaches working in Australia. It was predicted that more manageable levels of workload and more control over role autonomy would be positively related to higher levels of mental well-being. Overall, coaches in the present study generally had high levels of mental well-being. Findings from the adjusted analysis indicated that older and female coaches had higher levels of mental well-being while no differences were discerned for either coach setting or (weekly) coaching activity. Workload manageability and work-life control were both associated with mental well-being scores; coaches with a greater ability to manage their workload and with more control over decisions made in the workplace reported higher levels of mental well-being.

Mean mental well-being scores as measured by the WEMWBS for male coaches are 50.6 and 51.9 for female coaches, which compares favourably to the general Australian population (48.1) as measured by the Australian Psychological Society [10] national survey. These scores were also higher than those in similar professions, with mean scores of teachers recorded at 47.2 [11]. Reasons for this may be related to an increased sense of belonging for coaches as they work within teams and associations, leading to an increase in availability of social support. There must also be some source of motivation to continue within the coaching role and contribute to the development of participants.

The findings indicated that older coaches (i.e., those over 50 years of age) had better mental well-being than younger coaches. The suggestion here is that coaches are better able to manage workplace factors as they age. More coaching/life experience or less responsibilities in their personal life could be influencing factors. These coaches may also be more settled in their career and earned the trust of the organisations they work for (they also report more control over their work life and more manageable workloads). This corroborates previous research that reported emotional exhaustion can decrease with age [22]. An assumption based on general populations [18] is that these coaches are better at self-regulating against the numerous workplace factors experienced. There has been growing support for targeting the development of self-regulation as an effective means of managing stress [40], encouraging coaches to control their thinking, emotions and reactions. While coaches have identified benefits to employing self-regulation strategies, it is also noted that support and guidance is required for these strategies to be effective [41].

No differences were identified in mental well-being between levels of coaching (i.e., high performance or community), hours spent in the coaching environment, and number of years coaching. This contradicts Fletcher and Scott’s [42] proposition that coaches at higher levels are more susceptible to lower levels of mental well-being due to the increased number of stressors they experience. A suggestion for this is that all coaches perceive stressors, but these may come from differing sources. At the high-performance end, these may be more performance related; for the community coach, these may be related more to participant retention. Coaching literature has noted the multitude of stressors coaches can encounter in the sport workplace and being able to manage these is crucial for both mental well-being and performance [43].

The findings of this study have been adjusted to compensate for level of coaching, age and coaching experience, indicating that females in this study have higher levels of mental well-being. This could be attributed to these women having more adaptive coping strategies and that they self-regulate emotions in a more positive manner. While some previous research [26,44] suggests women in coaching roles experienced higher levels of stressors in the workplace, Durand-Bush et al.’s [41] concluded that female coaches use more proactive coping strategies to better manage these workplace factors. Considering the many roles coaches are required to undertake, these women may also be better at focusing on specific tasks and therefore reducing the impact of stressors. Norman [45] advocated for the education of female coaches to understand the context of coaching challenges and better prepare them for the workplace.

The findings indicated that workload was also related to mental well-being, with a more manageable workload associated with higher levels of coach mental well-being. Most coaches will have periods where their workload will be increased (i.e., tournament coaching), but being prepared for these is important for mental well-being. High work demands for long periods of time will have detrimental effects on coach mental well-being [46]. Organisational psychology literature [47] has identified the need for optimising workloads to improve mental well-being in the workplace. As such, organisations employing or caring for (i.e., coaching associations) are encouraged to develop guidelines and promote strategies to assist coaches in managing workloads. Many athlete associations have incorporated specific policy dictating the hours available for professional athletes to complete non-sport specific activities (i.e., team sponsor meetings), and this approach may be of benefit to high-performance coaches. At the community level, engaging a wider volunteer support group may help manage the coaching workload. Organisational psychology encourages monitoring of workplace stressors and providing employees the resources to cope and flourish [5]. Similarly, coaches themselves should recognise how they work most effectively and attempt to balance individual workloads as a means of maintaining peak performance. Kellmann, Altfeld and Mallett [48] highlighted the importance of the stress–recovery balance for coaches, particularly at a high-performance level. Likewise coach mental well-being can be affected by work–life balance discrepancies [49]. Didymus, Rumbold and Staff [50] recommend coaches take personal responsibility for their own mental well-being by ensuring they allocate time to non-coaching activities.

Organisational approaches to assist employees manage workload have typically focused on stress management interventions or training as a means to improve mental well-being [51]. This approach is also applicable to sports coaching, where it is not feasible to remove stressors from the workplace. Both relaxation and biofeedback strategies have been utilised to manage workplace stress, which are applicable to all levels of coaching and could become part of continued education programs to support coach retention. In other settings, cognitive–behavioural strategies are employed to support individuals. These approaches have been identified as having greater success in managing employee mental well-being than individuals taking personal responsibility [52]. The majority of research from sports coaching domains supports this [26,29,32], recommending organisations support coaches to balance workloads to prevent potential burnout. Again, this provides a call for associations to create policy to encourage the proactive management of work hours. All these approaches are focused on managing stressors, but there may be an opportunity for organisations to reduce the number of stressors experienced. Further research is required to establish whether this is possible within sports coaching.

This study also found a relationship between control (i.e., the capacity to influence decisions) and coaches’ mental well-being. Higher levels of control and autonomy in the workplace were associated with higher levels of mental well-being. Control in the sports coaching setting may include freedom to choose the wider coaching team and support staff, autonomy in athlete recruitment, and clear performance objectives. Organisational psychology literature has identified that having a limited ability to influence decisions and a lack of organisational support can have a negative impact on physical and mental well-being [53]. For sports coaches, autonomy in the role is a predictor of mental well-being [54], with a lack of control negatively influencing mental well-being [30]. Less control is also linked with emotional exhaustion [55] and decreased engagement [42].

Organisational support and an ability to exercise professional autonomy will be beneficial to improving the mental well-being of coaches [56]. Didymus, Rumbold and Staff [50] advocated for organisations to “ensure that the performance climate created by leaders provides coaches with the resources and entrustability (i.e., trusting coaches to perform their role with minimal supervision or micromanagement) to thrive under pressure” (p. 265). Further to this having clear expectations and role clarity is important for improved mental well-being [32]. Organisational leaders play a pivotal role here, by identifying clear performance objectives that are within the coach’s control and not based solely on athlete performance. This may also assist managing known stressors related to contract renewal and prolonged engagement [42].

A lack of control and having too many responsibilities are known stressors for coaches [57]. Similarly, having greater control in the workplace encourages the empowerment of employees [47]. For organisations, this implies a need to remove role ambiguity for coaches. In some cases, this may be difficult, as the numerous roles vary for coaches at different performance levels and work capacities (i.e., full time paid; volunteer). While the present study found no difference in mental well-being between coaches based on these factors, further research is required to ascertain how these may lead to perceived stress, and how coaches manage such stressors. This has been a focal point of organisational psychology for a number of decades, with the Job Demands—Control Model [58] central to this. The basic premise of the model is that high job demands and having minimal control create the most stressful workplaces. More recent research using this model found that employees with high levels of workplace control were able to better self-regulate against increased workloads, resulting in higher levels of satisfaction and lower anxiety [59].

Overall, the development of self-regulation practices should be a focus of a sub-set of individual coaches, the organisation employing them, and the association supporting them. Within coach education programs, at all levels, more emphasis should be placed on how the coach can perform at their best (including self-regulation strategies) and less on the specifics of what to coach. Having a greater understanding of the aspects associated with coaching may assist coaches to better manage these and their own mental well-being [60]. Coaches could benefit from working with an applied sport psychologist to help manage their own mental well-being [30]. The use of mindfulness techniques has also been recommended for coaches [61]. It is also important for organisations to consider coach mental well-being to reduce turnover rates [62]. Employers of coaches should target interventions for improving mental well-being [63], providing training to better assist the management of workplace factors [9].

The strengths of this study include the large sample size of coaches from a variety of sports within Australia, which is likely to be generalisable to coaches globally. While much research has investigated the impact of stress on coaches and implications for burnout, this study is one of the first to examine the level of mental well-being reported by coaches. The Areas of Work Life Scale has been widely used in organisational settings to examine work engagement and risk of burnout and holds promise as an instrument to assist the assessment of coach workplace factors and coach mental well-being. Future studies are encouraged to use this assessment tool to identify workload and control factors, and to explore how the other factors (reward, community, fairness, values) may influence coach mental well-being. Study limitations included survival bias—the fact that these coaches are currently active, with ex-coaches who have burnout not represented. This study is further limited by the lack of personal and social variables measured that are known to effect mental well-being (i.e., social support available). The inclusion of a specific measure of social support would be beneficial.

## 5. Conclusions

This study was one of the first to examine levels of mental well-being in Australian sports coaches. Responses from coaches surveyed identify that these sports coaches had generally high levels of mental well-being, supporting the first hypothesis. While this hypothesis postulated that coaches who are male, older, and have fewer weekly coaching hours have higher levels of mental well-being, the sub-group analysis identified younger and male coaches were the group most at risk of suffering lower levels of mental well-being. Of particular importance to all coaches is the ability to have a manageable workload and control over decisions made in the workplace, which supports the second hypothesis. The findings support the belief that self-regulation strategies to manage the effect of known and unknown stressors are important for sports coaches. Individual coaches are encouraged to take personal responsibility for their own mental well-being. However, a greater impact may be made by organisations employing or caring for coaches to create policy and best practice to ensure that coaches are supported in managing their mental well-being. This study highlights the role workplace factors, particularly workload and control, have on mental well-being and encourages future research to adopt the Areas of Work Life model to investigate this further. As a whole, sports coaches face a multitude of stressors and must manage these to ensure high levels of performance and mental well-being.

## Figures and Tables

**Table 1 ijerph-16-04601-t001:** Demographic characteristics of sample.

Variable	*n*	%
Gender		
Male	362	78.9
Female	97	21.1
Age group (years)		
≤30	58	12.5
31–50	290	62.5
>50	116	25.0
Coach setting		
Performance	128	27.6
Community	336	72.4
Coaching activity (h/week)		
≤10	188	40.5
>10–≤20	183	39.4
>20	93	20.0
Highest level coach accreditation		
None	17	3.8
Level 1	242	53.4
Level 2	138	30.5
Level 3	56	12.4
Period coaching main sport (years)		
≤2	97	20.9
>2–≤5	110	23.7
>5–≤10	106	22.8
>10	151	32.5
Coaching type		
Individual athlete(s) only	61	13.2
Team(s) only	318	68.8
Individual athlete(s) and team(s)	83	18.0

**Table 2 ijerph-16-04601-t002:** Summary of subgroup analyses for coach well-being (Warwick–Edinburgh Mental Well-Being Scale (WEMWBS) total score).

Variable	M (SD)	*t/F*	*p*	*d/η* ^2^
Demographic				
Gender				
Male	50.6 (7.7)	−1.41	0.16	−0.16
Female	51.9 (8.3)			
Age group (years)				
≤30	50.1 (8.8)	6.53	0.002	0.03
31–50	50.2 (7.6)			
>50	53.2 (7.3)			
Coaching				
Coach setting				
Performance	51.7 (8.3)	1.22	0.22	0.13
Community	50.7 (7.6)			
Coaching activity (h/week)				
≤10	51.3 (8.0)	0.95	0.39	0.03
>10–≤20	51.0 (7.5)			
>20	50.0 (8.0)			

**Table 3 ijerph-16-04601-t003:** Multivariate model predicting coach mental well-being (WEMWBS total score).

Variable	*b*	*se*	*p*
Gender (ref: female)	−1.70	0.83	0.04
Age (ref: >50 years			
≤30 years	−2.81	1.17	0.02
31–50 years	−2.70	0.79	0.001
Coach setting (ref: community)	1.48	0.81	0.07
Coaching activity (ref: ≤10 h/week)			
>10–≤20 h/week	−1.06	0.75	0.16
>20 h/week	−0.60	0.96	0.53
AWS workload	2.80	0.47	<0.001
AWS control	3.58	0.44	<0.001

Areas of Work Life Scale (AWS) workload/control subscales: 1 = strongly disagree, 5 = strongly agree.

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
