# Peer review of "Examining the Mental Well-Being of Australian Sport Coaches"

_ijerph, 2019, doi:10.3390/ijerph16234601_

Round 1

Reviewer 1 Report

The article includes a study on how the determinants of mental well-being can affect the coach both positively and negatively. The levels of mental well-being and their relationship with the workload and control are investigated. The results of the study indicate that the group with the highest risk of having lower levels of mental well-being was composed of young and male coachs.

Below are general and specific comments to improve the content and readability of the manuscript.

General comments:

The introduction provides sufficient background on the topic and previews major points. Both research design and analysis are adequate.

Specific comments:

2/72: Delete "a number of". 4/179; 5/192; 6/203: The format of the tables included in the results section should be standardized.

Author Response

Thank you for the positive comments.

Line 72 has been amended

Tables have been standardized and formatted correctly

Reviewer 2 Report

This study highlights the role workplace factors, particularly workload and control, have on mental well-being. The study is innovative and addresses important information on the area of coach mental well-being. I recommend its publication after minor changes.

Instrument and measures/ 136-144 Studies have indicated that the AWS is valid and has acceptable reliability [33, 34]. I recommend adding the exceptional reliability and validity of scale (AWS/ WEMWBS).

Procedure/ It is mandatory to inform the approval of the ethics committee; the approval number of the ethical permission is as much as possible

Discussion/Although the main finding has been correctly reported at the beginning of this section, a short remarking on the aim of the study could be reported just to provide more impact to the results of the study.

Materials and Methods/Did you realize a sample size estimation? If not, why?

Author Response

Thank you for the positive feedback.

Cronbach alpha values have been added to support this. (L. 141 and L.147) The ethical approval number has been included in the text. (L.157) Discussion sentence adjusted to term ‘aim’ and prediction sentence inserted to further clarify aims. (L.212-214).     Sample size response - A predictor to case ratio of 20:1 is recommended for MLR analyses (Tabachnik & Fidel, 2013) or minimum n of 200. Given the population large study population and anticipated response rate it was deemed that there would be no issue with obtaining the required sample size for the proposed analyses.

Reviewer 3 Report

Regarding the manuscript entitled "Examining the mental well-being of Australian sport coaches", I believe that the authors develop a work of interest, rigorous and that fits the special issue of the journal. Therefore, I think that this could be accepted after making some minor changes:

- It is recommended to briefly expand the results section of the summary including the findings of the predictive model that is developed.
- The introduction is correct and uses current references. In addition, the authors appropriately define the objective and hypotheses of their study.
- Perhaps the inclusion of this reference could enhance the theoretical framework of his work: Solstad, B. E., Ivarsson, A., Haug, E. M., & Ommundsen, Y. (2018). Youth sport coaches ’well-being across the season: The psychological costs and benefits of giving empowering and disempowering sports coaching to athletes. International Sport Coaching Journal, 5 (2), 124-135.
- The design of their scientific article is rigorous, in addition, an appropriate and relevant sample selection method is used, so I congratulate the authors for this.
- The explanation of the instrument is correct and detailed. The authors are requested to include the value of Cronbach's alpha to determine its internal consistency.
- They could detail more information in the procedure: confidentiality, presence of researchers, approval by ethics committee (include code).
- The analysis of the data and results are correct.
- The discussion is appropriate and relevant, in addition to following an appropriate discursive thread. Likewise, the main limitations and strengths of their study are indicated. Removing this work could help improve this information: Olusoga, P., Bentzen, M., & Kentta, G. (2019). Coach burnout: A scoping review. International sport coaching journal, 6 (1), 42-62.
- Authors must complete the conclusions by answering the level of compliance with the hypotheses.

Author Response

Thank you for the positive comments.

An additional sentence explaining the results has been added L.25-26 of abstract. Thank you for the suggestion to include an additional study. This study has been referenced in the text (L.77-78) Cronbach alpha values have been included (L. 141 and L.147) Further detail has been added within the procedures (L. 155-158) We note that the Olusoga et al. article wasn’t included in the original submission and thus we have not removed the article. If however, the reviewer was suggesting inclusion of the article then we could consider this adjustment if this was appropriate. The conclusion has been amended to address the hypotheses (L. 351 – 354)
